# Effect of Pharmacoprophylaxis on Postoperative Outcomes in Adult Elective Colorectal Surgery: A Multi-Center Retrospective Cohort Study within an Enhanced Recovery after Surgery Framework [note 1]

**DOI:** 10.3390/healthcare11233060

**Published:** 2023-11-28

**Authors:** William Olin Blair, Mary Allison Ellis, Maria Fada, Austin Allen Wiggins, Rachel C. Wolfe, Gourang P. Patel, Kara K. Brockhaus, Molly Droege, Laura M. Ebbitt, Brian Kramer, Eric Likar, Kerilyn Petrucci, Sapna Shah, Jerusha Taylor, Paula Bingham, Samuel Krabacher, Jin Hyung Moon, Monica Rogoz, Edson Jean-Jacques, Robert K. Cleary, Ransome Eke, Rachelle Findley, Richard H. Parrish

**Affiliations:** 1Department of Biomedical Sciences, School of Medicine, Mercer University, Columbus Campus, Columbus, GA 31902, USA; william.olin.blair@live.mercer.edu (W.O.B.); austin.allen.wiggins@live.mercer.edu (A.A.W.); jin.hyung.moon@live.mercer.edu (J.H.M.); jeanjacques_e@mercer.edu (E.J.-J.); eke_r@mercer.edu (R.E.); 2Department of Pharmacy, University of Kentucky Medical Center, Lexington, KY 40536, USA; mael229@uky.edu (M.A.E.); laura.means@uky.edu (L.M.E.); 3Heritage School of Osteopathic Medicine, Ohio University, Athens, OH 45701, USA; mf443720@ohio.edu; 4Department of Pharmacy Services, Barnes-Jewish Hospital, St. Louis, MO 63110, USA; rachel.wolfe@bjc.org; 5Department of Pharmacy Services, University of Chicago Hospitals, Chicago, IL 60637, USA; gourang.patel@uchospitals.edu (G.P.P.); kerilyn.petrucci@uchospitals.edu (K.P.); 6Department of Pharmacy Services and Surgery, Trinity Health Ann Arbor Hospital, Ann Arbor, MI 48104, USA; kara.brockhaus@trinity-health.org (K.K.B.); robert.cleary@trinity-health.org (R.K.C.); 7Department of Pharmacy Services, University of Cincinnati Medical Center, Cincinnati, OH 45219, USA; molly.droege@uchealth.com (M.D.); binghapr@mail.uc.edu (P.B.); samuel.krabacher@uchealth.com (S.K.); 8Department of Pharmacy Services, OhioHealth/Grant Medical Center, Columbus, OH 43215, USA; brian.kramer@ohiohealth.com; 9Department of Pharmacy Services, West Virginia University Medicine, Morgantown, WV 26506, USA; elikar@wvumedicine.org; 10Department of Pharmacy Services, Beaumont Hospital—Troy, Troy, MI 48085, USA; sapna.shah@beaumont.org; 11Department of Pharmacy Services, Legacy Good Samaritan Hospital, Portland, OR 97210, USA; jeetaylo@lhs.org (J.T.); mmrogoz@lhs.org (M.R.); 12Faculty of Medicine, Dalhousie University, Halifax, NS B3H 4R2, Canada; rachellef08@gmail.com

**Keywords:** enhanced recovery after surgery, ileus, outcomes, postoperative, pain, postoperative, pharmacotherapy, postoperative nausea and vomiting, prophylaxis, surgical site infection, surgery, colorectal, venous thromboembolism

## Abstract

Background: The application of enhanced recovery after surgery principles decreases postoperative complications (POCs), length of stay (LOS), and readmissions. Pharmacoprophylaxis decreases morbidity, but the effect of specific regimens on clinical outcomes is unclear. Methods and Materials: Records of 476 randomly selected adult patients who underwent elective colorectal surgeries (ECRS) at 10 US hospitals were abstracted. Primary outcomes were surgical site infection (SSI), venous thromboembolism (VTE), postoperative nausea and vomiting (PONV), pain, and ileus rates. Secondary outcomes included LOS and 7- and 30-day readmission rates. Results: POC rates were SSI (3.4%), VTE (1.5%), PONV (47.9%), pain (58.1%), and ileus (16.1%). Cefazolin 2 g/metronidazole 500 mg and ertapenem 1 g were associated with the shortest LOS; cefotetan 2 g and cefoxitin 2 g with the longest LOS. No SSI occurred with ertapenem and cefotetan. More Caucasians than Blacks received oral antibiotics before intravenous antibiotics without impact. Enoxaparin 40 mg subcutaneously daily was the most common inpatient and discharge VTE prophylaxis. All in-hospital VTEs occurred with unfractionated heparin. Most received rescue rather than around-the-clock antiemetics. Scopolamine patches, spinal opioids, and IV lidocaine continuous infusion were associated with lower PONV. Transversus abdominis plane block with long-acting local anesthetics, celecoxib, non-anesthetic ketamine bolus, ketorolac IV, lidocaine IV, and pregabalin were associated with lower in-hospital pain severity rates. Gabapentinoids and alvimopan were associated with lower ileus rates. Acetaminophen, alvimopan, famotidine, and lidocaine patches were associated with shorter LOS. Conclusions: Significant differences in pharmacotherapy regimens that may improve primary and secondary outcomes in ECRS were identified. In adult ECRS, cefotetan or ertapenem may be better regimens for preventing in-hospital SSI, while ertapenem or C/M may lead to shorter LOS. The value of OA to prevent SSI was not demonstrated. Inpatient enoxaparin, compared to UFH, may reduce VTE rates with a similar LOS. A minority of patients had a documented PONV risk assessment, and a majority used as-needed rather than around-the-clock strategies. Preoperative scopolamine patches continued postoperatively may lower PONV and PDNV severity and shorter LOS. Alvimopan may reduce ileus and shorten LOS. Anesthesia that includes TAP block, ketorolac IV, and pregabalin use may lead to reduced pain rates. Acetaminophen, alvimopan, famotidine, and lidocaine patches may shorten LOS. Given the challenges of pain management and the incidence of PONV/PDNV found in this study, additional studies should be conducted to determine optimal opioid-free anesthesia and the benefit of newer antiemetics on patient outcomes. Moreover, future research should identify latent pharmacotherapy variables that impact patient outcomes, correlate pertinent laboratory results, and examine the impact of order or care sets used for ECRS at study hospitals.

## 1. Introduction

The adoption of enhanced recovery after surgery (ERAS^®^) core items has been shown to decrease serious postoperative complications (POCs) and reduce postoperative hospital length of stay (LOS), readmission rates, and overall hospital costs [1,2,3,4,5,6,7,8,9]. Protocolized prophylaxis of common POCs with pharmacotherapy, such as surgical site infection (SSI) with appropriate antibiotics [10,11,12,13], venous thromboembolism (VTE) with anticoagulants [14,15,16,17], and postoperative nausea and vomiting (PONV) using multi-modal approaches [18,19,20,21,22,23], among others, coupled with preoperative risk assessments [24,25,26,27,28,29,30,31,32,33,34], has become more prevalent in surgical practice. Still, elucidation of pharmacotherapy regimens associated with lower POCs, LOS, and readmission remains somewhat nascent [5,13,16,35,36]. Moreover, estimates of the impact of pharmacotherapy prophylaxis on the attainment of positive patient outcomes in hospitals with “homegrown” enhanced recovery audit and feedback systems are publicly non-existent unless reported on public-facing hospital webpages, which, according to some reports, may be misleading information [37,38]. 

In our study, the effect of pharmacotherapy prophylaxis to prevent SSI, VTE, PONV, pain (POP), and ileus (POI) (primary outcomes) as well as to reduce LOS and readmission rates (secondary outcomes) is presented in five parts. Part 1 presents the multi-center methodology and descriptive statistics for pharmacotherapy and procedural-related variables. Antibiotics and SSI prophylaxis are found in Part 2. In Part 3, anticoagulants and VTE prophylaxis are highlighted. Part 4 deals with antiemetics and PONV prevention. Finally, Part 5 describes analgesics and adjunctive agents, and POP and POI impact. Opportunities for improvement of these primary and secondary outcomes using pharmacoprophylaxis are identified and highlighted in the discussion.

## 2. Materials and Methods

Clinical pharmacists, pharmacy residents, and medical and pharmacy students on clinical rotations from 10 hospitals in the United States participated in the study. The clinical pharmacists were contacted and recruited (by Dr. Parrish) through electronic listservs of the American College of Clinical Pharmacy Perioperative Care and Critical Care Practice and Research Networks. Patients at each site were identified through a search of ICD-10 and CPT codes in electronic health records (EHR) pertaining to colorectal diagnoses and procedures (Appendix A). Each hospital underwent a study initiation training, and each data collector was trained in chart abstraction and entry procedures in REDCap^®^ that included a total of 144 pharmacotherapy-related variables per patient [39]. 

This multi-center retrospective cohort study is based on a research strategy for ERAS^®^-related pharmacotherapy prophylaxis introduced in a previous report [40] that was trialed in a single-center randomized cohort study for elective colorectal and gynecological oncology surgery patients [5]. The methodology and data dictionary of the present study were described previously and modified through several iterations by all co-investigators [41]. A simple randomization algorithm (http://www.random.org) was used at each site to select up to 50 patients ≥18 years of age admitted as an inpatient for elective colorectal surgery between 1 January 2021 and 31 December 2021 for inclusion in the analysis. The Caprini score was used for estimating VTE risk [17], Apfel scoring for PONV risk [18], and a calculator on Global RPh was used to estimate oral morphine milligram equivalents (MME) [42]. Each of these calculators was embedded in a REDCap^®^ data collection tool (Appendix A). To detect a significant difference for the least common POC, venous thromboembolism (VTE), with a 0.2 β and 80% power and an α of <0.05, a total sample size of at least 565 complete cases was needed based on an estimated 4% and 1.9% VTE incidence in the population and cohort, respectively. However, an interim analysis of the VTE rate showed that the actual VTE rate in 476 cases was 1.5%. Therefore, a sample size of 378 cases would be needed for statistical significance, and data collection was stopped.

Descriptive statistics were used for frequency tabulations, χ^2^ for cross-tabulations of categorical variables, and linear and logistic regression were employed to measure associations for colorectal surgeries between dependent (medication-related) and independent (outcome-related) variables. Categorical variables were presented as N (%), and continuous variables were presented as mean (±S.D.) or median (IQR). Univariate analyses were conducted to evaluate differences in baseline patient characteristics, operative characteristics, and postoperative outcomes between patients who received antibiotics, anticoagulants, and antiemetics prior to and after surgery. Regression analysis using a PICO-styled research question series was used to determine the effect of various medication regimens on composite primary and secondary outcomes. Outcomes included the frequency and severity of POCs as assessed using Clavien–Dindo classification [43], LOS in days, and 7- and 30-day readmission. All statistical analyses were conducted using Stata Statistical Software: Version 17, 2021. College Station, TX, USA: StataCorp LLC., and data have been reported in line with the STROCSS criteria (Appendix B) [44]. The Institutional Review Boards of all hospitals reviewed and exempted the study from further review (Mercer University IRB #H2201008; all other site-related IRB determination letters and data use agreements are on file). This study was conducted in accordance with the Declaration of Helsinki and is registered at Research Registry (unique identifying number researchregistry7683) at https://www.researchregistry.com/browse-the-registry#home/registrationdetails/62192040a30665001e1d8cef/ (accessed on 15 October 2023).

## 3. Results

### 3.1. Part 1—Descriptive Statistics

Data from 476 adult patients who underwent an elective colorectal procedure at 10 sites were included in the analysis. These hospitals comprised a total of 6716 beds (R = 210–1274 beds/hospital) and performed a total of 4522 (452 ± 208 mean procedures/hospital) colorectal surgeries during calendar year 2021. Most of these participating hospitals collect colorectal surgery data internally through either service-based or institutional review processes (n = 8), and seven provide their perioperative care teams with feedback on program performance. In addition to colorectal procedures at participating hospitals, other common specialty procedures included orthopedics and gynecology (90%), urological (70%), and obstetrics and spine (60%).

Table 1 shows patient baseline characteristics. Slightly more of the patients were female, and most were white. The average weight was 82 ± 21 kg, and the mean eGFR was 80.9 ± 24.91 mL/min/1.73 m^2^. In the group of patients that regularly consumed alcoholic beverages, most had between 1 and 7 drinks per week. Almost one-half had a drug allergy, with most of those having a non-penicillin/non-cephalosporin allergy. A cancer diagnosis was found in about half of patients, and over two-thirds did not receive neoadjuvant radiation or chemotherapy. The median LOS was 4 days (IQR = 3–6.25 days); the mean LOS was 5.6 ± 4.9 days (LOS range: 1–58 days). Readmission rates within 7- and 30-days were 6% and 7.8%, respectively. POCs, LOS, readmission, and other historical and drug-related variables were not significantly different for baseline characteristics.

Table 2 describes the procedure-related variables. The two most common colorectal resection sites were single site or combinations of sigmoid colon and rectum. Most patients underwent a laparoscopic procedure, followed by open and robotic laparoscopy. The most frequently employed anesthetic combinations were general inhalation, propofol, and IV short-acting opioids such as fentanyl or remifentanil. The typical blood loss volume was 119 mL, and over 60% of patients had an ASA score of III. Almost all patients received lactated Ringer’s solution, and over 25% received albumin. Peripheral intravenous fluids were generally stopped by POD 3 in almost three-quarters of cases. Within the preceding 6 months, 13.4% of patients received iron therapy with oral iron (sulfate and gluconate) as the predominant medication. Insulin was administered in about one-fourth of cases, and most of those received it on a sliding scale.

A breakdown of surgical technique by anatomical location, LOS, and readmission rates is found in Appendix A. The sigmoid colon was the most common anatomical location, followed by the rectum, ascending colon, and descending colon. The appendix and cecum were the least common locations. The most common surgical techniques were laparoscopic and open, with non-robotic techniques predominating over robotic 4 to 1. Laparoscopic procedures had the shortest LOS, with manual procedures having the highest 7-day readmission rate and robotic having the highest 30-day readmission rate as compared to open procedures. Open manual procedures had the lowest 7-day and similar 30-day readmission rates compared to laparoscopic. Converted to open from laparoscopic had the highest 30-day readmission rate and a comparable LOS to open manual. Sigmoid colon was the most common procedure of all procedures, and the rectum was the most frequent anatomical location for laparoscopic procedures.

The relationships between anatomical location and LOS and readmission rates are presented in Appendix A. Procedures that included the appendix had the longest LOS and the lowest 7- and 30-day readmission rates of all procedures and combinations. Procedures including the cecum had the shortest LOS. Procedures including the transverse colon and small intestine had the highest 7-day readmission rates, and those including the descending colon and sigmoid had the highest 30-day readmission rates. For LOS, procedures involving the transverse, small intestine, and rectum had significant variability. There was no difference in 7- and 30-day readmission for any colonic location.

### 3.2. Part 2—Antibiotics and SSI

Table 3 reports prophylactic antibiotic utilization. SSI during hospitalization was diagnosed in 3.4% of patients. In-hospital infection was diagnosed in 7.4% of patients as well as in 8.5% of patients at a recent post-discharge visit. Intravenous antibiotic (IVA) prophylaxis was administered in almost all cases, and cefazolin 2 g/metronidazole 500 mg (C/M) was the most common IVA prophylaxis regimen. Cefoxitin 2 g was the second-most used IVA, followed by ertapenem 1 g. Mechanical bowel preparation (MBP) was prescribed in over two-thirds, with laxatives and oral antibiotic bowel prep (OABP) prescribed in most cases (a three-dose regimen of each metronidazole 500 mg and neomycin 1 g). Almost 60% received both OABP and IVA to prevent SSI. More Caucasians than Blacks/African-Americans received both OABP/IVA. OABP/IVA was almost twice as likely in robotic surgeries without reductions in SSI, LOS, and readmission. One-fourth of cases received postoperative IVA, and the majority that received IVA postoperatively had a duration of greater than 4 days. The majority received their first prophylactic dose within 30 min of surgical incision. Overall, over one-third received an intraoperative re-dose. Over three-quarters of cases received chlorhexidine skin preparation preoperatively.

Table 4 shows the most frequently used intravenous antibiotics ranked by primary and secondary outcomes. LOS for C/M and ertapenem was significantly different from that for the second-generation cephalosporins, cefotetan and cefoxitin. C/M was the most frequently used antibiotic, followed by ertapenem and cefoxitin. C/M use was associated with the shortest average LOS and 7-day readmission rate but higher than average in-hospital infection rate and 30-day readmission rate. Ertapenem also was associated with the shortest average LOS and lower than average in-hospital infection rate but higher than average post-discharge infection and 7- and 30-day readmission rates. Of note, cefotetan use was associated with no in-hospital infections, with longer-than-average LOS and post-discharge infection rates. Combination ampicillin/cefoxitin (n = 39) was associated with the lowest rates of in-hospital and post-discharge infection and 30-day readmission rates but with higher than average 7-day readmission. Cefoxitin had a longer LOS compared to other antibiotics. Patients with a penicillin or cephalosporin allergy seemed to stay in the hospital longer but had lower-than-average rates for primary and secondary outcomes.

Small intestine procedures that included other colonic segments had an SSI rate that was over twice that of small intestine-only procedures and almost 3 times the overall SSI average rate (Appendix A). Large intestine procedures had the lowest SSI rate, which occurred at half the average SSI rate. C/M had an overall SSI rate of 12.9% for procedures involving the small intestine. Cefotetan and ertapenem use had no SSIs for any colonic procedures. The SSI rate for large colonic procedures was significantly lower than that for those involving the small intestine (*p* = 0.00182).

### 3.3. Part 3—Anticoagulants and VTE

Anticoagulation regimens are found in Table 5. The incidence of any in-hospital VTE was 1.5%. Patients were VTE risk-stratified preoperatively in about one-half of cases, and the average Caprini score indicated a high risk for post-surgical VTE. Preoperative VTE pharmacoprophylaxis was administered in over three-fourths of cases, and unfractionated heparin (UFH) 5000 units SC was given in almost all cases within 6 h of incision. Sequential compression devices were employed in three-fourths of cases. Anticoagulation was continued postoperatively in 96.0%, and only 26.9% were prescribed for at-home use. Enoxaparin 40 mg SC daily was the most common postoperative in-hospital and at-home anticoagulation regimen. Most patients who were prescribed at-home antithrombosis management received anticoagulation between 3 and 4 weeks. Of the 210 patients with a cancer diagnosis, only 45.7% (n = 96) received at-home VTE prophylaxis, and the most frequently prescribed anticoagulants were enoxaparin 40 mg SC daily and apixaban 2.5 mg PO twice daily. Several hospitals administered preoperative UFH after epidural placement and then switched to enoxaparin postoperatively.

Table 6 shows a breakdown of pharmacologic anticoagulation by VTE, LOS, bleeding/hematoma, and readmission rates. Enoxaparin 40 mg SC daily was the most frequently used anticoagulant regimen in-hospital and post-discharge and was associated with the shortest LOS and rates of VTE and 30-day readmission compared to UFH. All VTEs occurred while on a regimen including UFH 5000 units SC q8h (one in sequential combination with enoxaparin 40 mg SC daily) for a combined VTE rate of 4.5% (7/156). Ketorolac IV use was not related to in-hospital or discharge bleeding or hematoma.

A comparison between enoxaparin and UFH is shown in Table 7. LOS and VTEs were significantly lower for enoxaparin use than for UFH use. There was no difference in in-hospital or discharge bleeding or hematoma and 7- or 30-day readmission.

Table 8 shows the at-home anticoagulation. Enoxaparin 40 mg SC daily was the most frequent anticoagulant, with the lowest VTE rate and 30-day readmission (*p* = 0.661). Apixaban 2.5 mg PO twice daily had the lowest complication and 7-day readmission rates. Patients on anticoagulation at home were more likely to be readmitted at 7 and 30 days. The bleeding rate at home while on an anticoagulant was 1.6%, and differences between the agents were significant (*p* = 0.004). Discharge anticoagulation was associated with a slightly higher 7- (*p* = 0.401) and 30-day (*p* = 0.661) readmission.

### 3.4. Part 4—Antiemetics and PONV/PDNV

Table 9 outlines the prophylaxis and treatment of PONV. Risk assessment for PONV was used in 40.3% of cases, and the typical risk percentage using Apfel scoring was between 0 and 20%. However, PONV occurred in 47.9% of cases. Those who received antiemetics prior to induction were usually administered dexamethasone, ondansetron, and scopolamine patches with one dose within 60 min of surgical incision. Similarly, for those receiving antiemetics at extubation, the most common agents were ondansetron and dexamethasone. Many patients received only one postoperative antiemetic dose (n = 82; 27.5%); however, 45 patients (51.1%) received greater than 7 doses during hospitalization. Over 70% were prescribed with an as-needed frequency, and the most common rescue agent was IV ondansetron. PONV incidence was approximately 30% in each of the following phases of care: PACU, first 12 h on the ward, and 12–24 h on the ward/unit, while 58.7% experienced PONV > 24 h on the ward and/or unit.

In Table 10, antiemetics and their effects on PONV and post-discharge nausea and vomiting (PDNV) are presented. No combination of antiemetics was associated with a lower PONV rate. Scopolamine patches, used in 13.4% of cases, were associated with significantly lower PONV and PDNV rates. The use of spinal opioids and IV lidocaine was associated with lower PONV (*p* < 0.05). PIVs containing saline and ondansetron were associated with lower PDNV (*p* < 0.05). Higher postoperative incremental MME doses increased the likelihood of PONV by 23% (*p* < 0.0001). Promethazine was associated with a higher 7-day readmission rate (*p* < 0.01). Prochlorperazine was associated with significantly higher rates of PONV and PDNV (*p* < 0.05).

Table 11 illustrates the variety of agents that may have a positive impact on pain, PONV, and/or ileus incidence. The most common agents in this category were propofol, gabapentinoids, alvimopan, and sub-anesthetic ketamine bolus. Magnesium sulfate was used for pain management in 75 cases (15.8%). For neuromuscular blockade reversal, sugammadex (n = 268; 56.3%) was predominant over neostigmine (n = 152; 31.9%).

### 3.5. Part 5—Analgesics and POP and POI

Multi-modal pain management is described in Table 12. Oral acetaminophen, often contained in a combination product with hydrocodone or oxycodone, was the most common analgesic. Acetaminophen was usually given every 6 h around the clock. IV acetaminophen was used in 17% of cases. A variety of non-specific COX and COX-2 non-steroidal anti-inflammatory agents were used, including ketorolac PO and IV, ibuprofen PO and IV, celecoxib, and naproxen. Gabapentin, ketamine IV, and ketorolac IV were the most common adjunctive agents administered for postoperative pain management. For opioid exposure, patients received a median of 42 (IQR: 25–88) oral MME pre- and intraoperatively and 67.5 (IQR: 22.5–180.75) MME postoperatively.

Table 13 reports the analgesics, anesthetics, and adjunctive agents associated with POP, POI, MME, LOS, and readmission. Transversus abdominis plane (TAP) block with long-acting local anesthetics, lidocaine IV, ketorolac IV, pregabalin, and celecoxib were associated with a lower pain rate. The most frequently used adjunctive agents with a potentially positive impact on ileus were gabapentinoids and alvimopan. Alvimopan was associated with a lower ileus rate (*p* < 0.001). Postoperative acetaminophen, both PO and IV, alvimopan, and lidocaine patches were associated with shorter LOS, and no agents were associated with lower readmission.

Not surprisingly, POP was the most common in-hospital POC, followed by PONV and delayed gastric emptying/ileus Appendix A. Although two patients expired, most POCs were minor and required only appropriate pharmacotherapy or management, except for in-hospital ileus (combined Clavien-Dindo grades IIIa and IIIb − n = 12; 2.5%). Pain and infection were the most frequent at-home POCs (Appendix A). POCs occurring at home were reportedly more severe (15.1% vs. 5.2%); over one-half of patients had a POC in the post-discharge phase.

Univariate and bivariate analyses related to the effect of pharmacoprophylaxis on patient postoperative outcomes (Appendix C) are summarized in Table 14 using a general PICO-styled methodology that has been utilized recently in the creation of various perioperative guidelines [45,46,47,48].

## 4. Discussion

### 4.1. Part 1—Overview

To our knowledge, this is the first comprehensive study to characterize the scope and measure the impact of pharmacotherapy prophylaxis regimens on common POCs, LOS, and readmission in a real-world multi-center cohort of adult ECRS patients. The impact of individual agents on primary and secondary outcomes, in many cases, was found to be mixed; some improved POC severity rates but adversely affected LOS and readmission rates and vice versa. Each major POC introduced in the preceding parts will be discussed in terms of the preferred agents used to prevent it with a comparison to existing literature similarities and unique findings.

### 4.2. Part 2—Antibiotics and Infection (SSI and Other)

The SSI rate found in this study (3.4%) is in the range of the typical rate for ECRS, 2 to 10% in North America and Europe. Each hospital used a different protocolized antibiotic regimen to prevent SSI, and over 10% continued IVA greater than 4 doses, unlike the ERAS^®^ Society recommended postoperative duration [1]. This variation in the primary antibiotic used allows a comparison to identify potential regimens of choice associated with better outcomes. Of the SSIs that occurred, 7 out of 16 patients received C/M, and 6 received cefoxitin alone. For ECRS, ASHP/SHEA/IDSA/SIS guidelines recommend the use of IV first-generation cephalosporin (cefazolin is the only parenteral first-generation available on North American markets) with metronidazole (C/M) or second-generation cephalosporins (cefotetan, cefoxitin, cefuroxime) as regimens of choice [35]. Cefuroxime was not used in our study. However, these guidelines are over 10 years old and include ertapenem at the end of the colorectal recommendations. Moreover, a 2008 report suggested that ertapenem use led to lower SSI rates and shorter LOS with a reported cost savings of over USD 2000 [49]. Another comparative study found that LOS from C/M use was not different than for cefotetan, but overall hospital costs were significantly higher [50]. Our study found that ertapenem and cefotetan provided better overall outcomes when compared to C/M or cefoxitin alone for ECRS [51,52]. Further, evidence suggests that ertapenem may be more effective than cefotetan [53]. Other more recent reports have shown similar results for ertapenem as ours, and this may be especially important in patients who are carriers of community- or hospital-acquired extended-spectrum β-lactamase-producing Enterobacteriaceae [54,55]. Now that ertapenem is available generically, guidelines and protocols for SSI prevention may require a re-examination.

One interesting finding was the use of the combination of ampicillin with cefoxitin. With an average LOS, use of this combination led to only one SSI (total rate 2.6%, ileocolectomy rate 11.1%; 1/9 and colectomy rate 0%; 0/30), with low rates of 7- and 30-day readmission. While we did not collect information about the microorganisms causing SSI or infection, the combination may provide better coverage against Enterococcus spp. The combination has not been studied systematically for prophylaxis in any surgical setting, but ampicillin could be added to regimens lacking enterococcal coverage as recommended for infectious treatment [56]. Indeed, significantly more SSIs occurred when the small intestine was operated on, but immunocompromise due to Crohn’s, ulcerative colitis, cancer treatments, or COVID-19 was not measured.

The SSI rate for patients receiving IVA and OA with MBP versus IVA alone was not different, in contrast to a recent network meta-analysis including over 12,000 patients conducted by Koo and colleagues [57]. However, they did not report SSI rates, only reduced odds of SSI. Further, MBP is known to cause clinically significant fluid and electrolyte imbalances throughout the perioperative period and, combined with OA, can disrupt normal gut microbiome for months, even years [58]. In our study, MBP/IVA/OA did not lead to shortened LOS, reduced SSI, or lower re-admission. To achieve reduced SSI rates, the use of MBP/IVA/OA, perhaps limited to laparoscopic and robotic ECRS cases as currently recommended, must be weighed against the risk of fecal contamination during the anastomosis. Tissue handling and exposure are better during laparoscopic and robotic cases when patients are bowel prepped, making the conduct of the operation easier and perhaps safer, even with the potential of generating multi-drug resistant organisms. Disruptions to the gut microbiome caused by OA, as well as from exposure of the bowel to oxygen and transient interruptions of local blood flow, can shift the constituents of the lumen towards obligate and facultative anaerobes. This shift can lead to increased infection, anastomotic leak, dysmotility, and malabsorption and may increase cancer risk and occurrence [59]. Of note, the type of surgical technique (robotic versus open) and patients’ race were significant determinants of the choice of antibiotic prophylaxis. It is unclear why white people received OA more often than black people.

Preoperative anemia is believed to be associated with higher SSI rates and represents an area for research [60]. Oral iron therapy initiated within 6 months of surgery to raise hemoglobin levels was associated with higher in-hospital infection. Parenteral iron was used rarely. One explanation may be that patients may have initiated oral iron too close to the procedure to make a difference in preoperative hemoglobin levels. However, hemoglobin levels and iron monitoring parameters were not collected. No type of insulin therapy in-hospital was associated with lower SSI or post-discharge infection.

### 4.3. Part 3—Anticoagulants and Venous Thromboembolism

VTE occurred in 1.5% of study participants, which is similar to previous estimates of 1.1% to 2.5% in large databases of postoperative patients with colon cancer and inflammatory bowel disease [61]. Enoxaparin 40 mg SC daily was the predominant pharmacoprophylaxis regimen, which aligns with the National Comprehensive Cancer Network recommendations [62]. However, the recent American Society of Hematology guideline does not differentiate between LMWH and UFH [63], and UFH was inferior to enoxaparin-containing regimens. Additionally, a sequential regimen of UFH followed by enoxaparin after epidural removal was associated with the longest LOS. This might also be because epidurals are more often used for open cases and not for laparoscopic/robot cases, and open cases are associated with longer LOS [64]. A recent comparison of enoxaparin versus UFH in a large dataset of abdominal surgery patients similarly found that enoxaparin may be associated with fewer in-hospital VTEs with similar profiles for VTE and major bleeding at 90 days [65]. Enoxaparin was the only LMWH used in our study, and extrapolation to other LMWHs used in other parts of the world should be avoided. Possible explanations for the differences between enoxaparin and UFH may be the occurrence of missing doses because of UFH’s multiple daily dosing schedule or a less predictable pharmacodynamic profile. Missing doses of UFH, however, were not collected.

Apixaban 2.5 mg PO twice daily was the second most used post-discharge anti-coagulant and seemed to perform well regarding bleeding/hematoma and 7-day readmission, although oral DOAC regimens generally had higher 30-day readmission rates compared to enoxaparin due to bleeding or hematoma. A recent post-hoc analysis of the AVERT trial validated the safety and efficacy of apixaban in surgical patients with gastrointestinal cancers [66]. Currently indicated as prophylaxis in orthopedic procedures, it is very likely that DOAC regimens will be incorporated into future VTE prophylaxis for colorectal surgeries [67]. In general, ERAS^®^ promotes earlier use of the enteral route for medication, fluid, and nutrient administration, and some subpopulations of surgical patients could be more satisfied with an oral versus injectable regimen [68]. The post-discharge prophylaxis rate for cancer patients in our study (45.7%) was lower than a recent survey of colorectal surgeons who reported using extended VTE prophylaxis in their colorectal cancer patients (54%). Therefore, adherence to recommendations still appears to be an opportunity [69].

### 4.4. Part 4—Antiemetics and Postoperative/Post-Discharge Nausea and Vomiting

Postoperative nausea and vomiting (PONV) were the second most common POC within our cohort. Rates were lower than those reported in a similar previous study; however, this difference may be accounted for by a higher proportion of females in the earlier study even though around-the-clock IV metoclopramide 48 h was used postoperatively in gynecological oncology patients [5]. Centers with and without participation in an ERAS^®^ protocol were included in the cohort. Use of a risk stratification system is likely less in centers without an ERAS^®^ protocol, and PONV stratification was lower than expected (40.3% in our study) [19]. This may contribute to higher PONV severity rates [70]. A large percentage of patients in the study experienced PONV for more than 24 h postoperatively. More prevention needs to be implemented immediately in the extended time window after surgery, as late PONV has a profound effect on LOS [23]. Moreover, none of the centers used newer antiemetics such as amisulpride, and few used palonosetron or any dose form of aprepitant within their prophylaxis or treatment strategies. This lack of use may reflect concerns about cost as well as potentiation of QTc prolongation, even though these newer antiemetics have been shown to be safer and more effective than older agents, especially in combination [71].

Another concern raised by this study is that of PONV treatment modalities. With nearly half of the cohort’s patients experiencing nausea or vomiting at some time postoperatively, most patients were prescribed as-needed antiemetic regimens, commonly IV ondansetron, after receiving IV dexamethasone as their preoperative prophylaxis. None reported using sub-hypnotic propofol doses for PONV which would be expected since this practice is primarily anesthesia-based in the PACU. This pattern is consistent with the frequent occurrence of PONV for more than 24 h postoperatively that is seen in our results. It seems the current practice of many healthcare systems is not to continue postoperative antiemetics and provide rescue therapies.

The cohort of patients in this study did not receive a particularly diverse range of antiemetic drugs; however, a large majority received some form of PONV prophylaxis. Expectedly, patients receiving prophylaxis had lower PONV rates than those who did not. IV prochlorperazine was associated with the worst PONV and PDNV rates, which may be explained because its use was primarily for rescue, not prophylaxis. Another older antiemetic, IV promethazine, was associated with a higher 7-day readmission rate, perhaps due to extravasation (occurring between 0.1 and 6% of administrations) [72], although these data were not collected. The use of a scopolamine patch yielded lower PONV and PDNV severity rates as well as shorter LOS and seems to be an effective and noninvasive treatment option for surgical patients [20]. Patients in this study who required higher MME saw a marked increase in their likelihood of PONV. On the other hand, patients receiving continuous IV lidocaine infusion as part of a multi-modal pain strategy with or without limited postoperative MME had lower PONV severity rates [73,74,75]. Administration of either albumin and/or transfused packed red blood cells in the hospital was associated with higher PDNV severity and may represent a new finding.

### 4.5. Part 5—Analgesics and Pain/Ileus (Delayed Gastric Emptying)

Multi-modal pain management has been advocated since the initial development of ERAS^®^ protocols; however, optimal combinations have not yet been determined. The combination of medications is novel to this study and represents a real-world implementation of the ERAS^®^ recommendations with the utilization of the Clavien–Dindo classification [43], which allows for a graded approach to pain severity assessment as opposed to a subjective scale.

Multiple medications utilized intraoperatively have been hypothesized to decrease postoperative pain severity and, subsequently, ileus severity. However, the discussion is still ongoing. Our study supports Sarakatsianou and colleagues, who found that continuous IV lidocaine infusion (n = 54) decreases both in-hospital and post-discharge pain severity as well as LOS; however, it is associated with a higher post-discharge ileus severity rate, which is somewhat conflicting [76]. Continuous IV magnesium infusion (n = 74) was also associated with higher in-hospital ileus severity; however, the benefit to in-hospital or post-discharge pain severity was not apparent in our study, unlike the results found by Ng and colleagues [77]. Injectable, non-intravenous medications such as TAP blocks were found to have lower in-hospital and post-discharge pain severity; however, spinal injections of local anesthetics with or without opioids were found to have higher in-hospital and post-discharge pain severity, although it is unclear why this would occur. Wound infiltration in our study was found to have lower severity rates of in-hospital ileus only for nonliposomal bupivacaine with epinephrine (n = 27). Conversely, liposomal bupivacaine (n = 17) was associated with higher rates of 30-day readmittance in our study, which complements the results found by Hussain et al. that perineural liposomal bupivacaine was not superior to nonliposomal bupivacaine [78].

Analgesics utilized postoperatively to decrease pain incidence and severity were found to have interesting trends in our study. Regarding the efficacy of IV versus PO acetaminophen, 81 patients received acetaminophen IV during their admission, and 446 received acetaminophen PO, both of which were associated with higher pain severity but shorter LOS, which could be a function of the limited time frame that patients receive IV acetaminophen at most institutions [79,80]. Some patients (n = 51) received both IV and PO acetaminophen during their stay, which could confound meaningful analysis of differential pain impact based on route. Non-steroidal anti-inflammatories (NSAIDs) produced variable outcomes in this study. Used around the clock, celecoxib and ketorolac IV were found to have lower pain severity rates, and ketorolac IV was the only one found to lower ileus severity rates. Ibuprofen IV was associated with higher MME use and higher in-hospital pain and ileus severity rates, most likely because it was used as needed in many cases, unlike celecoxib. While NSAIDs can decrease POP and POI, there is a potential risk of anastomotic leak, which might lead the surgery team to avoid NSAIDs in particular patients. As adjunctive agents utilized for pain control, lidocaine patches and methocarbamol had a negative linear relationship and may have been initiated after the occurrence of pain or ileus rather than prophylactically.

The utilization of a multi-modal approach not only assisted with decreased pain but also decreased ileus severity rate secondary to decreased utilization of opioids. Alvimopan use was associated with decreased MME, in-hospital pain severity, and in-hospital and post-discharge ileus severity. However, the use of alvimopan likely resulted in a decrease in in-hospital pain severity secondary to reducing ileus symptoms rather than a direct effect on pain itself, as has been hypothesized. Moreover, while alvimopan carries a black box warning for myocardial infarction, limiting its use to a 15-dose maximum, it may not be appropriate for all patients [81]. Interestingly, of the medications that decreased pain and ileus severity rates, pregabalin and TAP block were associated with a longer LOS. This LOS impact may be because these medications may have been utilized on patients who were perceived to have a higher POC risk and subsequently needed to stay admitted longer for additional monitoring.

### 4.6. Strengths and Limitations

This study has several strengths. It is a real-world cohort study conducted in the same period at 10 hospitals across the US. In addition, a randomized selection process occurred with each center’s coordinator using the same random number generator to identify ECRS cases, data collectors were trained in live and recorded educational sessions, and the number of cases needed to demonstrate statistical significance for the least common POC was calculated prior to and during the study. Hospital site coordinators were provided with the results from their sites to help address current practices to improve overall clinical outcomes and surgical quality metrics. Finally, procedures involving the small intestine and appendix, not stratified in most colorectal studies, were included in this study and illustrated significant differences in SSI rates as compared to large colon-only procedures.

On the other hand, the study has a few limitations due to its retrospective nature, which might preclude cause and effect assignment; however, medication administration to prevent a POC preceded the occurrence of any POC reported. Selection bias cannot be ruled out as a potential cause of error. Data collection and chart abstraction may have had variations due to the recording of end-point variables at different institutions within the medical record. It is possible that some emergent cases were included in case selection even though the definition of an elective case was the documentation of a preadmission visit. Another limitation of this multi-center study is that each center had its own protocols, and therefore, medications, doses, and durations may differ from site to site. However, this heterogeneity potentially allows for broader generalizability to a real-world, diverse ECRS patient population. It does, however, mean that some medications may have utility in ECRS patients, but the utilization was too low to be able to generate meaningful data. Additional limitations to pain control analysis include that, for patient-controlled analgesia (PCA) at one institution, MME calculations were estimated using milligram amounts of opioid dispensed to rather than administered by the patient and may represent an overestimation of MME utilization. No laboratory measurements nor the time course of the administration of any medication (other than pre-, intra-, and post-operatively) were collected, which might provide a more in-depth explanation of associations. Doses for medications other than for SSI and VTE were not collected, and the frequency of administration for pain management pharmacotherapy was often not reported. This study included only a few non-pharmacologic modalities (postoperative ambulation, use of sequential compression devices, and aromatherapy for PONV), and it is known that other bundled perioperative interventions, such as shorter preoperative LOS, avoiding surgical drains, and early removal of urinary catheters, have a major impact on operative throughput and POC [81]. In addition, these results pertain only to adult elective colorectal cases, and extrapolation to other surgical procedures in adults or any peri-procedure in children should be avoided. While our results are largely descriptive and associative from univariate and bivariate analyses, an additional manuscript employing a conceptual framework using R software, version 4.3.2 to uncover latent interactional pharmacotherapy variables in a secondary analysis is underway [82].

## 5. Conclusions

Significant differences in pharmacoprophylaxis outcomes for common POCs, LOS, and readmission rates, among others, were identified. In adult ECRS, cefotetan or ertapenem may be better regimens for preventing in-hospital SSI, while ertapenem or C/M may lead to shorter LOS. The value of OA to prevent SSI was not demonstrated. Inpatient enoxaparin, compared to UFH, may reduce VTE rates with a similar LOS. A minority of patients had a documented PONV risk assessment, and a majority used as-needed rather than around-the-clock strategies. Preoperative scopolamine patches continued postoperatively may lower PONV and PDNV severity and shorter LOS. Alvimopan may reduce ileus and shorten LOS. Anesthesia that includes TAP block, ketorolac IV, and pregabalin use may lead to reduced pain rate. Acetaminophen, alvimopan, famotidine, and lidocaine patches may shorten LOS. Given the challenges of pain management and the incidence of PONV/PDNV found in this study, additional studies should be conducted to determine optimal opioid-free anesthesia and the benefit of newer antiemetics on patient outcomes. Moreover, future research should identify latent pharmacotherapy variables that impact patient outcomes, correlate pertinent laboratory results, and examine the order or care sets for study hospitals.

## Figures and Tables

**Table 1 healthcare-11-03060-t001:** Patient baseline characteristics by LOS as a comparison.

**Age (yrs.)**	Female	59.4 (±14.7)
	Male	56.2 (±16.0)
**Sex (n; %)**	Female	247 (51.9)
	Male	229 (48.1)
**Race (n; %)**	Asian	12 (2.5)
	Native Hawaiian or other Pacific Islander	1 (0.2)
	Black or African-American	59 (12.4)
	White	382 (80.3)
	Hispanic	6 (1.3)
	Unknown/Not reported	16 (3.3)
**Weight (kg)**	82 ± 21	
**Preoperative eGFR (mL/min/1.73 m^2^)**	80.9 ± 24.91	
**Ethanol history/week**	None	329 (72.5)
	1–7 drinks	109 (24.0)
	8–14 drinks	9 (2.0)
	Greater than 14 drinks	7 (1.5)
**Documented drug allergy**	None	255 (53.6)
	Non-penicillin/non-cephalosporin	177 (80.1)
	Penicillin	60 (27.1)
	Cephalosporin	9 (4.1)
**Preoperative cancer diagnosis**	Yes	210 (44.1)
	No	266 (55.9)
**Neoadjuvant therapy**	None	144 (68.6)
	Chemotherapy	64 (30.5)
	Radiation	34 (16.2)

**Table 2 healthcare-11-03060-t002:** Intestinal and procedure-related variables.

Procedure-Related Variable (n; %)		
**Intestinal segment** **(includes multiple colonic segments)**	Sigmoid colon	242 (50.8)
	Rectum	150 (31.5)
	Ascending colon (including hepatic flexure)	146 (30.7)
	Descending colon (including splenic flexure)	145 (30.5)
	Small intestine	122 (25.6)
	Transverse colon	119 (25.0)
	Cecum	75 (15.8)
	Appendix	16 (3.4)
**Surgical technique**	Laparoscopic	243 (51.0)
	Open	137 (28.8)
	Robotic	96 (20.2)
**American Society of** **Anesthesiologists (ASA) score**	I	2 (0.4)
	II	167 (35.1)
	III	289 (60.7)
	IV	18 (3.8)
**Estimated blood loss during surgery** **(mL ± S.D.)**	119.6 ± 190.2	

**Table 3 healthcare-11-03060-t003:** Prophylactic anti-infective use variables.

**In-hospital SSI**	16 (3.4)	
**In-hospital infections**	35 (7.4)	
**Post-discharge** **infections**	40 (8.5)	
**Pre-incisional IVA** **administered?**	Yes	467 (98.1)
**More than 1 IVA** **administered?**	Yes	246 (52.7)
**Intraoperative IVA** **re-dose administered?**	Yes	177 (37.2)
**First (or only) IVA** **administered**	Cefazolin 2 g	124 (26.6)
	Cefoxitin 2 g	106 (22.7)
	Ertapenem 1 g	80 (17.1)
	Metronidazole 500 mg	62 (13.3)
	Cefotetan 2 g	52 (11.1)
	Ampicillin 1 g	9 (1.9)
	Piperacillin/tazobactam 3.375 g	4 (0.8)
	Ceftriaxone 2 g	3 (0.6)
	Piperacillin/tazobactam 4.5 g	2 (0.4)
	Ampicillin/sulbactam 3 g	1 (0.2)
	Vancomycin 1 g	1 (0.2)
**Second IVA** **administered**	Metronidazole 500 mg	126 (51.6)
	Cefazolin 2 g	49 (20.1)
	Ampicillin 1 g	30 (12.2)
**Combination IVA** **administered**	Cefazolin 2 g/metronidazole 500 mg	158 (33.2)
	Cefoxitin 2 g/ampicillin 1 g	39 (8.2)
	Metronidazole 500 mg/gentamicin 5 mg/kg	9 (1.9)
	Clindamycin 600 mg/gentamicin 5 mg/kg	5 (1.0)
	Cefazolin combinations (other)	5 (1.0)
	Ciprofloxacin 400 mg/metronidazole 500 mg	3 (0.6)
	Cefotetan 2 g/metronidazole 500 mg	1 (0.2)
	Levofloxacin 500 mg/metronidazole 500 mg	1 (0.2)
**Postoperative IVA** **administered?**	Yes	98 (24.1)
**Duration of** **postoperative IVA**	1 dose	7 (7.1)
	2 doses	12 (12.1)
	3 doses	14 (14.3)
	4 doses	21 (21.4)
	>4 doses	44 (44.9)
**Timing of IVA prior to** **incision**	0–15 min	158 (33.8)
	16–30 min	170 (36.4)
	31–45 min	68 (14.6)
	46–60 min	31 (6.6)
	>60 min	35 (7.5)
**Skin preparation** **administered**	Chlorhexidine	382 (80.3)
	Povidone-iodine	104 (21.8)
	None	15 (3.2)

**Table 4 healthcare-11-03060-t004:** The overall most frequently used intravenous antibiotics, LOS, and infection rates, 7- and 30-day readmission.

Antibiotic (N of Patients Treated— Doses in Table 3)	Ave LOS	Hospital Infection Rate (%)	Discharge Infection Rate (%)	7-Day Readmit Rate (%)	30-Day Readmit Rate (%)
C/M (158)	5.0	9.7	7.1	2.6	10.3
ertapenem (80)	5.0	5.0	11.4	8.9	8.0
cefoxitin (76)	7.1	10.4	6.5	5.2	6.5
cefotetan (52)	5.9	0	11.5	5.8	7.7
ampicillin/cefoxitin (39)	5.7	3.0	6.0	9.1	6.0
*Sub-total (405)*	*5.6*	*7.1*	*8.3*	*5.3*	*8.3*
** *Penicillin or cephalosporin allergic patients* **					
metronidazole/ gentamicin (9)	7.7	0	0	0	11.1
metronidazole (5)	2.0	0	0	0	0
clindamycin/ gentamicin (5)	5.5	20.0	0	20.0	0
levofloxacin (4)	6.0	0	25.0	0	0
ciprofloxacin/ metronidazole (3)	4.7	0	33.3	0	0
vancomycin (1)	6	0	0	0	0
*Sub-total (27)*	*6.7*	*3.7*	*7.4*	*3.7*	*3.7*
** *Miscellaneous beta-lactams and combinations* **					
cefazolin (7)	6.3	14.3	28.6	28.6	14.3
piperacillin/tazobactam (6)	11.5	16.7	16.7	16.7	0
cefazolin combinations (other) (5)	4.6	0	0	0	0
ceftriaxone (3)	5.7	33.3	0	0	0
ampicillin/sulbactam (1)	11	0	0	0	0
cefotetan/ metronidazole (1)	2	0	0	0	0
*Sub-total (23)*	*7.2*	*13.0*	*13.0*	*13.0*	*0*
*None (21)*	*3.7*	*0*	*0*	*0*	*0*
**Totals (476)**	5.5	6.7	8.0	5.3	7.4

**Table 5 healthcare-11-03060-t005:** Venous thromboembolism (VTE) prophylaxis use variables.

**Preoperative anticoagulant administered?**	Yes	368 (77.3)
	UFH 5000 units SC w/in 6 h	364 (98.9)
**Postoperative in-hospital anticoagulant administered?**	Yes	453 (95,2)
	Enoxaparin 40 mg SC daily	262 (57.8)
	UFH 5000 units SC q8h	125 (27.6)
	UFH 5000 units SC q8h followed by enoxaparin 40 mg SC daily	31 (6.8)
	UFH 5000 units SC q12h	21 (4.6)
	Enoxaparin 40 mg SC q12h	7 (1.5)
	UFH 5000 units SC q8h followed by enoxaparin 40 mg SC q12h	4 (0.9)
	UFH 5000 units SC q12h followed by enoxaparin 40 mg SC daily	3 (0.7)
**N of postoperative in-hospital doses administered**	0	1 (0.2)
	1	25 (5.5)
	2	72 (15.8)
	3	73 (16.0)
	4	45 (9.9)
	5	32 (7.0)
	6	31 (6.8)
	7	25 (5.5)
	>7 *	152 (33.3)
**Non-pharmacologic VTE prophylaxis in hospital (multiple methods included)**	None	18 (3.8)
	Ambulation	274 (57.6)
	Compression stockings	4 (0.8)
	Sequential compression device	364 (76.5)
**Post-discharge at-home anticoagulant given?**	Yes	128 (26.9)
	Enoxaparin 40 mg SC daily	83 (64.8)
	Apixaban 2.5 mg PO twice daily	29 (22.7)
	Rivaroxaban 10 mg PO daily	11 (8.6)
	Warfarin daily (various doses)	5 (3.9)
**Days postoperative at-home anticoagulant**	1–7 days	2 (1.6)
	8–14 days	8 (6.4)
	15–21 days	30 (21.0)
	22–28 days	54 (43.2)
	>28 days	31 (24.8)
**Cancer patients receiving at-home VTE prophylaxis (96/210)**	Enoxaparin 40 mg SC daily	64 (66.7)
	Apixaban 2.5 mg PO twice daily	16 (16.7)
	Warfarin PO daily at various doses	5 (5.2)
	Apixaban 5 mg PO twice daily	4 (4.2)
	Rivaroxaban 20 mg PO daily	4 (4.2)
	Rivaroxaban 15 mg PO daily	2 (2.0)
	Rivaroxaban 10 mg PO daily	1 (1.0)
**Anticoagulant dose adjusted based on weight**	Yes	7 (1.2)
**Anticoagulant dose adjusted based on renal function**	Yes	6 (1.6)

* mostly because of q12h and q8h dosing.

**Table 6 healthcare-11-03060-t006:** The most frequently used anticoagulant regimens, LOS, and numbers and rates of VTE, in-hospital and discharge bleeding/hematoma, and readmission.

Anticoagulant Regimen (Frequency of Use)	Ave LOS (d)	VTE Rate (%)	Hospital Bleeding/Hematoma Rate %	Discharge Bleeding/Hematoma Rate (%)	7-Day Readmit Rate (%)	30-Day Readmit Rate (%)
enoxaparin 40 mg SC daily (262)	5.1	0	9.5	2.7	4.9	7.2
heparin 5000 units SC q8h (125)	5.8	4.8	4.8	0.8	5.6	8.8
heparin 5000 units q8h/ enoxaparin 40 mg daily (31)	8.5	3.2	6.4	0	12.9	12.9
heparin 5000 units SC q12h (21)	6.5	0	4.2	4.2	4.2	8.4
enoxaparin 40 mg SC q12h (7)	11.8	0	14.3	0	0	0
heparin 5000 units q8h/ enoxaparin 40 mg q12h (4)	23	0	25	0	0	0
heparin 5000 units q12h/ enoxaparin 40 mg SC daily (3)	7	0	0	0	0	0
Totals (453)	5.7	1.5	7.9	1.9	5.5	7.9

**Table 7 healthcare-11-03060-t007:** Comparison of enoxaparin and unfractionated heparin use for VTE, bleeding/hematoma, LOS, and readmission.

Anticoagulant Administered (n Includes Sequential Dual Agent Therapy)	Significance (*p*-Values)					
	LOS	VTE	In-Hospital Bleeding/Hematoma	Discharge Bleeding/Hematoma	7-Day Readmit	30-Day Readmit
**enoxaparin 40 mg SC (307) compared to UFH 5000 units SC (184)**	<0.0001	0.004	0.190	0.398	0.834	0.613

**Table 8 healthcare-11-03060-t008:** At-home anticoagulation use and discharge bleeding/hematoma and readmit rates.

Anticoagulant Regimen	Ave Days of Home Therapy	Discharge Bleeding/Hematoma Rate (%)	7-Day Readmit Rate (%)	30-Day Readmit Rate (%)
enoxaparin 40 mg SC daily (83)	15–22	1.2	8.4	9.6
apixaban 2.5 mg BID (22)	22–28	0	0	13.6
rivaroxaban 20 mg daily (11)	>28	18.1	0	18.1
apixaban 5 mg BID (7)	>28	0	14.2	14.2
warfarin daily (5)	>28	0	20	0
Totals (128)		1.6	7.0	10.9

**Table 9 healthcare-11-03060-t009:** Postoperative nausea and vomiting (PONV) prophylaxis use variables.

**Preoperative/intraoperative antiemetic?**	Yes	417 (87.6)
**Antiemetics given prior to induction (includes multiple agents)**	Dexamethasone IV	186 (44.6)
	Ondansetron IV	84 (20.1)
	Scopolamine patch	64 (15.3)
	Aprepitant PO	4 (1.0)
	Prochlorperazine IV	3 (0.7)
	Metoclopramide IV	2 (0.5)
	Promethazine IV	2 (0.5)
	Perphenazine PO	1 (0.2)
**Antiemetic given prior to extubation (298)**	Yes	234 (78.5)
	Ondansetron	216 (72.5)
	Dexamethasone	100 (33.6)
	Scopolamine patch	12 (4.0)
	Promethazine	5 (1.7)
	Metoclopramide	4 (1.3)
	Prochlorperazine	3 (1.0)
	Aprepitant PO	1 (0.3)
**Postoperative antiemetic (PACU/ward) for rescue**	Yes	310 (65.1)
	Ondansetron IV	251 (81.0)
	Ondansetron PO	92 (29.7)
	Promethazine IV	75 (24.2)
	Prochlorperazine IV	59 (19.0)
	Promethazine PO	40 (12.9)
	Prochlorperazine PO	16 (5.2)
	Metoclopramide IV	7 (2.3)
	Metoclopramide PO	5 (1.6)
	Palonosetron	2 (0.6)
**Number of rescue antiemetic doses** **administered postoperatively (excluding aprepitant and scopolamine patch)**	None	48 (16.1)
	1 dose	82 (27.3)
	2 doses	51 (17.1)
	3 doses	32 (10.7)
	4 doses	14 (4,7)
	5 doses	11 (3.7)
	6 doses	11 (3.7)
	7 doses	4 (1.3)
	>7 doses	45 (15.1)
**Postoperative nausea and vomiting (PONV)**	Yes	228 (47.9)
**PONV time of occurrence**	In PACU	68 (30.5)
	<12 h on ward	69 (30.9)
	12–24 h on ward	68 (30.5)
	>24 h on ward	131 (58.7)

**Table 10 healthcare-11-03060-t010:** Antiemetics, anesthetics, and IV fluids administered by PONV and PDNV.

Medication or IV Fluid	*p* Value
Lower PONV	
Scopolamine	<0.001
Lidocaine IV	<0.05
Spinal opioid	<0.05
Higher PONV	
Prochlorperazine	<0.05
Albumin	<0.05
Lower PDNV	
Ondansetron	<0.05
Saline-containing IV	<0.05
Scopolamine	<0.05
Higher PDNV	
Spinal opioid	<0.01
Packed red blood	<0.05
Albumin	<0.05
Prochlorperazine	<0.05

**Table 11 healthcare-11-03060-t011:** Pharmacotherapy that can affect PONV.

**Administered during hospitalization**	Propofol	436 (91.6)
	Sugammadex	268 (56.3)
	Alvimopan	227 (47.7)
	Gabapentin	212 (44.5)
	Famotidine IV	158 (33.2)
	Ketamine IV analgesia bolus	157 (33.0)
	Neostigmine	152 (31.9)
	Acetaminophen IV	97 (20.4)
	Pregabalin	78 (16.4)
	Magnesium sulfate IV for pain	75 (15.8)
	Dexmedetomidine	65 (13.7)
	Ketamine IV continuous	10 (2.1)

**Table 12 healthcare-11-03060-t012:** Pharmacotherapy that can affect POP/POI.

**Intraoperative anesthesia (includes multiple types) (n; %)**	Gaseous general	459 (96.4)
	Propofol	379 (79.4)
	Short-acting opioid (fentanyl, remifentanil)	319 (67.0)
	Midazolam	248 (52.1)
	TAP block w/long-acting local anesthetics	114 (23.9)
	Dexmedetomidine	65 (13.7)
	Epidural	63 (13.2)
	Lidocaine continuous IV	55 (11.6)
	Wound infiltration w/non- liposomal bupivacaine without epinephrine	28 (5.9)
	Wound infiltration w/non- liposomal bupivacaine with epinephrine	27 (5.7)
	Spinal opioid and LA	25 (5.3)
	Wound infiltration w/liposomal bupivacaine only	17 (3.6)
	Spinal opioid	12 (2.5)
**Nonopioids (includes** **multiple agents) (n; %)**	Acetaminophen PO	446 (93.7)
	Gabapentin	176 (37.0)
	Ketamine IV analgesia bolus	157 (33.0)
	Ketorolac IV	143 (30.0)
	Methocarbamol	93 (19.5)
	Acetaminophen IV	81 (17.0)
	Lidocaine 5% patch	79 (16.6)
	Ibuprofen PO	75 (15.8)
	Magnesium sulfate IV for pain	75 (15.8)
	Celecoxib	58 (12.2)
	Ibuprofen IV	25 (5.3)
	Pregabalin	24 (5.0)
	Naproxen	14 (2.9)
	Ketamine IV continuous	10 (2.1)
	Ketorolac PO	4 (0.9)
	Meloxicam PO	3 (0.7)
**Additional agents administered during hospitalization (n; %)**	Sugammadex	268 (56.3)
	Alvimopan	227 (47.7)
	Famotidine IV	158 (33.2)
	Neostigmine	152 (31.9)

**Table 13 healthcare-11-03060-t013:** Anesthetics, analgesics, and adjunctive agents by pain, ileus, morphine milligram equivalents (MME), LOS, and readmission.

Medication or IV Fluid	*p*-Value
**Lower in-hospital pain**	
TAP block w/long-acting local anesthetics	<0.001
Lidocaine IV	<0.001
Alvimopan	<0.001
Ketorolac IV	<0.001
Pregabalin	<0.001
Celecoxib	<0.01
Ketamine non-anesthetic bolus	<0.05
Propofol	<0.05
Midazolam	<0.05
Famotidine	<0.05
**Higher in-hospital pain**	
Lidocaine patch	<0.001
Ibuprofen IV	<0.001
Methocarbamol	<0.001
Acetaminophen PO	<0.01
Acetaminophen IV	<0.05
Ibuprofen PO	<0.05
**Lower post-discharge pain**	
Short-acting opioid (fentanyl, remifentanil)	<0.001
TAP block w/long-acting local anesthetics	<0.001
Propofol	<0.001
Alvimopan	<0.001
Lidocaine IV	<0.01
Famotidine	<0.05
Ketamine bolus	<0.05
Midazolam	<0.05
Pregabalin	<0.05
**Higher post-discharge pain**	
Spinal opioid with local anesthetics	<0.001
Spinal opioid	<0.01
Gabapentin	<0.01
Acetaminophen PO	<0.01
**Lower ileus**	
Alvimopan	<0.001
Ketorolac IV	<0.01
Gabapentin	<0.01
Midazolam	<0.01
TAP block w/long-acting local anesthetics	<0.05
Wound infiltration w/non-liposomal bupivacaine w/epinephrine	<0.05
**Higher ileus**	
Ibuprofen IV	<0.001
Magnesium sulfate IV	<0.05
**Lower post-discharge ileus**	
None	
**Higher post-discharge ileus**	
Lidocaine IV	<0.05
**Less than 50 MME intraoperative**	
None	
**Higher than 50 MME intraoperative**	
Ibuprofen IV	<0.01
**Less than 50 MME postoperative**	
Alvimopan	<0.01
**Higher than 50 MME postoperative**	
Dexmedetomidine	<0.01
Magnesium sulfate	<0.05
Sugammadex	<0.05
**Shorter LOS**	
Acetaminophen PO	<0.01
Alvimopan	<0.01
Acetaminophen IV	<0.01
Lidocaine patch	<0.01
Famotidine	<0.05
**Longer LOS**	
Pregabalin	<0.01
TAP block w/long-acting local anesthetic	<0.05
**Lower 7-day readmission**	
None	
**Higher 7-day readmission**	
Promethazine	<0.01
Gabapentin	<0.05
**Lower 30-day readmission**	
None	
**Higher 30-day readmission**	
Ketamine continuous infusion	<0.01
Dexmedetomidine	<0.05
Wound infiltration w/liposomal bupivacaine	<0.05

**Table 14 healthcare-11-03060-t014:** PICO-style questions with recommendations for medication use to address POCs, LOS, and readmission.

PICO Question—In Elective Colorectal Surgery (ECRS):	Recommendation Summary
**Antibiotics and surgical site infection (SSI)**	
**1.** **Which IV antibiotic(s) (IVA) or combination of IVA and oral antibiotics (OA) is (are) associated with a lower incidence of in-hospital or post-discharge SSI?**	There is no relationship between OA and SSI, in-hospital infection, or post-discharge infection.
**2.** **Is preoperative iron therapy (ferrous sulfate, ferrous gluconate, iron dextran, ferric derisomaltose, ferric carboxymaltose, ferric gluconate, ferumoxytol, iron sucrose) associated with a lower incidence of in-hospital infection?**	Preoperative use of oral iron products is associated with a higher incidence of in-hospital infection (*p* < 0.05).
**3.** **Is in-hospital insulin therapy (regular sliding scale, regular by basal-bolus correction, insulin glargine, insulin detemir, NPH insulin, premixed insulin 70/30, premixed insulin 75/25, regular insulin infusion) associated with a lower incidence of in-hospital infection?**	No type of insulin therapy in-hospital was associated with lower SSI or post-discharge infection. Use of NPH insulin was associated with higher post-discharge infections (*p* < 0.05).
**4.** **Which preoperative IVA(s) (ampicillin ampicillin/sulbactam, cefazolin, cefotetan, cefoxitin, ceftriaxone, ciprofloxacin, clindamycin, ertapenem, gentamicin, levofloxacin, metronidazole, piperacillin/tazobactam, vancomycin) is (are) associated with a lower incidence of in-hospital infection?**	Lower in-hospital infection will occur when either cefotetan or ertapenem are used (*p* < 0.05). Cefoxitin and C/M use were associated with the highest SSI rates (6/106; 5,7% and 7/158; 4.4%, respectively).
**5.** **Which preoperative IVA(s) is (are) associated with a lower incidence of post-discharge infection?**	Lower post-discharge infection will occur when cefazolin is used (*p* < 0.05).
**6.** **Which preoperative IVA(s) is (are) associated with a shorter LOS or lower 7- or 30-day readmission?**	Longer LOS when cefoxitin or piperacillin/tazobactam are used (*p* < 0.01).
**Anticoagulants and venous thromboembolism (VTE)**	
**7.** **Which anticoagulant(s) is (are) associated with a lower incidence of in-hospital VTE?**	Enoxaparin 40 mg subcutaneously (SC) daily was associated with lower in-hospital VTE incidence (OR: 11.3; 95% CI: 1.36–95.25; *p* = 0.025). All VTE events occurred when unfractionated heparin (UFH) 5000 units SC q8h (UFH) was ordered (n = 7; 3.8%; *p* = 0.004).
**8.** **Which anticoagulant(s) is (are) associated with a lower incidence of in-hospital bleeding/hematoma?**	There was no difference between enoxaparin and UFH regimens for in-hospital bleeding (*p* = 0.19).
**9.** **Which anticoagulant(s) is (are) associated with a shorter LOS?**	Average LOS for enoxaparin (5.1 days) and UFH (5.9 days) alone were significantly shorter than for sequential UFH (q8h or q12h) and enoxaparin (daily or q12h (9.7 days) (*p* = 0.004).
**10.** **Which anticoagulant(s) is (are) associated with reduced readmission?**	There was no difference between enoxaparin and UFH regimens for 7-day (*p* = 0.83) and 30-day readmission (*p* = 0.61).
**Antiemetics and postoperative/post-discharge nausea and vomiting**	
**11.** **Which antiemetic(s) and IV fluid(s) is (are) associated with lower PONV/PDNV?**	Lower PONV (*p* = 0.001) and PDNV (*p* < 0.05) will occur when a scopolamine patch is used. Lower PONV will occur when lidocaine IV (*p* < 0.05) is used. Lower PDNV will occur when ondansetron and 0.9% NaCl-containing IV infusion are used (*p* < 0.05). Higher PONV and PDNV will occur when prochlorperazine and albumin are used (*p* < 0.05). Higher PDNV will occur when packed red cells are used (*p* < 0.05).
**12.** **Which antiemetic(s) is (are) associated with shorter LOS or readmission?**	No antiemetic was associated with lower 7- or 30-day readmission. Shorter LOS will occur when preoperative famotidine (*p* < 0.05) is used. Lower 7-day readmission will occur when promethazine is not used (*p* < 0.05).
**13.** **Which anesthetic agent(s) is (are) associated with lower PONV/PDNV?**	Lower PONV (*p* < 0.05) but higher PDNV (*p* < 0.01) will occur when spinal opioids are used.
**14.** **Which analgesic(s) and adjunctive agent(s) is (are) associated with lower PONV/PDNV?**	None are associated with lower PONV or PDNV.
**Analgesics, anesthetics, and adjunctive agents and pain and ileus**	
**15.** **Which anesthesia type(s) is (are) associated with lower ileus?**	Lower ileus will occur when midazolam (*p* < 0.01) and TAP block with long-acting anesthetics and wound infiltration with non-liposomal bupivacaine w/epinephrine (*p* < 0.05) are used and when lidocaine IV is not used (*p* < 0.05).
**16.** **Which analgesic(s) (acetaminophen IV, acetaminophen oral, celecoxib, ibuprofen IV, ibuprofen oral, ketorolac IV, lidocaine IV, lidocaine patch, naproxen) and adjunctive pain agents (alvimopan, dexmedetomidine, gabapentin, ketamine bolus, ketamine infusion, magnesium sulfate IV, methocarbamol, neostigmine, pregabalin, propofol, sugammadex) or combinations is (are) associated with lower total morphine milligram equivalents (MME)?**	Less than 50 mg intraoperative MME were administered when ibuprofen IV (*p* < 0.01) was not used. Less than 50 mg postoperative MME were administered when alvimopan (*p* < 0.01) was used and when dexmedetomidine (*p* < 0.01), and when magnesium sulfate IV and sugammadex (*p* < 0.05) were not used.
**17.** **Which analgesic(s) and adjunctive pain agents is (are) associated with lower in-hospital pain?**	Lower in-hospital pain complication when celecoxib, lidocaine IV, ketorolac IV, TAP block with long-acting local anesthetics, and pregabalin (*p* < 0.001), celecoxib (*p* < 0.01), ketamine non-anesthetic bolus, and propofol are used (*p* < 0.05). Lower pain complication when ibuprofen IV, lidocaine patch, methocarbamol (*p* < 0.001), acetaminophen PO (*p* < 0.01), and acetaminophen IV and ibuprofen PO (*p* < 0.05) are used.
**18.** **Which analgesic(s) and adjunctive pain agents is (are) associated with lower in-hospital and post-discharge ileus?**	None are associated with lower post-discharge ileus; in fact, lower post-discharge ileus when lidocaine IV (*p* < 0.05) was not used. Lower in-hospital ileus when alvimopan (*p* < 0.001), ketorolac IV, and gabapentin (*p* < 0.01) are used. Lower in-hospital ileus when ibuprofen IV (*p* < 0.001) and magnesium sulfate IV for pain (*p* < 0.05) are not used.
**19.** **Which analgesic(s) and adjunctive pain agents is (are) associated with shorter LOS or readmission?**	Shorter LOS when acetaminophen PO and IV, alvimopan, and lidocaine patch (*p* < 0.01) and famotidine (*p* < 0.05) are used. Pregabalin was associated with longer LOS (*p* < 0.01). Lower 7-day readmit when gabapentin is not used (*p* < 0.05).
**20.** **Which anesthesia type(s) is (are) associated with a shorter LOS or readmission?**	Shorter LOS when TAP block with long-acting local anesthetic (*p* < 0.05) was not used. Lower 30-day readmit when ketamine continuous infusion (*p* < 0.01), liposomal bupivacaine, and dexmedetomidine (*p* <0.05) are not used.
**Procedure-related effects**	
**21.** **Which anatomical surgical site location(s) is (are) associated with a shorter LOS or lower readmission?**	Procedures that included the appendix had the highest LOS and the lowest 7- and 30-day readmission rates of all procedures and combinations. Procedures including the cecum had the lowest LOS. Procedures including the transverse colon and small intestine had the highest 7-day readmission rates, and those including the descending colon and sigmoid had the highest 30-day readmission rates. For LOS, procedures involving the transverse, small intestine, and rectum had significant variability. There was no difference in 7- and 30-day readmission for any colonic location.
**22.** **Which surgical technique(s) is (are) associated with a shorter LOS or lower readmission?**	Laparoscopic procedures had the lowest LOS, with manual procedures having the highest 7-day readmission rate and robotic having the highest 30-day readmission rate as compared to open procedures. Open manual procedures had the lowest 7-day and similar 30-day readmission rates compared to laparoscopic. Converted to open from laparoscopic had the highest 30-day readmission rate and a comparable LOS to open manual.

## Data Availability

Due to limitations in data sharing agreements, data generated in this study cannot be shared.

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
