# Peer review of "Effect of Pharmacoprophylaxis on Postoperative Outcomes in Adult Elective Colorectal Surgery: A Multi-Center Retrospective Cohort Study within an Enhanced Recovery after Surgery Frameworkâ€"

_healthcare, 2023, doi:10.3390/healthcare11233060_

Round 1
Reviewer 1 Report
Comments and Suggestions for Authors
The article is wery well writed. The colected data are significant and the aim of the study was achieved. In the extensive discussion that was held, a current topic that deserves to be highlighted was omitted: the role of the intestinal microbiome. I think a mention of this issue would make the article even better. With this small change I think it can be published
Author Response
We are grateful for the reviewer's kind comments and helpful suggestions.

Reviewer 2 Report
Comments and Suggestions for Authors
The manuscript “Effect of pharmacoprophylaxis on postoperative outcomes in adult elective colorectal surgery: A multi-center retrospective cohort study within an enhanced recovery after surgery frame-work”. The study aims to show pharmacotherapy regimens that improve primary and secondary outcomes in ECRS. There are few suggestions given below that should be addressed to improve the quality of the manuscript.
1. If no significance was obtained for table 1 then authors can remove the p value column from table 1 and add a sentence in text and figure legend that “no significance was obtained….”
2. Did authors find any correlation between the drugs used in the tables?
Author Response

(The authors gave the same response as above.)

Reviewer 3 Report
Comments and Suggestions for Authors
Healthcare-2701432
In this study titled “Effect of pharmacoprophylaxis on postoperative outcomes in 2 adult elective colorectal surgery: A multi-center retrospective 3 cohort study within an enhanced recovery after surgery frame-4 work” by Blair et al., the authors aimed to find out the impact of different pharmacotherapy on postoperative recovery and complications in surgical patients. Study has been designed in a scientifically robust method involving appropriate statistics. However, there are a few minor concerns that may help in improvisation of the manuscript.
Authors mentioned about surgical site infection (SSI) of 3.4%.
Were these 3.4% patients having any history of immune-complications or any other complications that might have been under-reported?
Please mention about shortest length of stay (LOS) and longest length of stay in numbers.
I recommend authors to state the conclusions of the study in the abstract very clear and concise. Just stating significant differences in pharmacotherapy regimens…… is not enough. What were the parameter that authors think as an important finding of the study should be leveraged in conclusion. Additionally, just saying future research should identify latent …………… is not giving any idea on the challenges for the future. Please provide a line or two on future research.
In the methods section authors state that “The present study derives from a research strategy for ERAS®-related pharmacotherapy prophylaxis introduced in a previous report and trialed in a single-center randomized cohort study for elective colorectal and gynecological oncology patients” but, in the abstract author states that “Records of 476 randomly selected adult patients who underwent elective colorectal surgeries (ECRS) at 10 US hospitals were abstracted”. This is a little confusing, I request authors to double check. If this is a single-center study, authors can state that as a limitation of the study.
Comments on the Quality of English LanguageMinor English language editing is required.
Author Response

(The authors gave the same response as above.)

Reviewer 4 Report
Comments and Suggestions for Authors
Thank you for letting me review this paper. The research is very interesting, however, some points have to be clarified:
- what criteria have been used to choose the antibiotic (for example piperacillin/tazobactam and ertapenem instead of cefazolin/metronidazole)? Do the hospitals have specific protocols?
- what criteria have been used to decide if the patient needed or not an oral antibiotic preparation?
- what criteria have been used to decide which drug has to be used for the antithrombotic prophylaxis (enoxaparin 40 mg SC daily and apixaban 2.5 mg PO twice daily)?
Thank you
Author Response

(The authors gave the same response as above.)
